# Declining Radial Growth in Major Western Carpathian Tree Species: Insights from Three Decades of Temperate Forest Monitoring

**DOI:** 10.3390/plants12244081

**Published:** 2023-12-06

**Authors:** Jergus Rybar, Zuzana Sitková, Peter Marcis, Pavel Pavlenda, Jozef Pajtík

**Affiliations:** 1National Forest Centre, Forest Research Institute, T.G. Masaryka 22, 960 01 Zvolen, Slovakia; zuzana.sitkova@nlcsk.org (Z.S.); peter.marcis@nlcsk.org (P.M.); pavel.pavlenda@nlcsk.org (P.P.); jozef.pajtik@nlcsk.org (J.P.); 2Faculty of Forestry, Technical University in Zvolen, T.G. Masaryka 24, 96001 Zvolen, Slovakia

**Keywords:** resilience, adaptation, drought, production, ICP Forests

## Abstract

This study investigates the radial growth response of five key European forest tree species, i.e., *Fagus sylvatica, Picea abies*, *Abies alba*, *Quercus petraea*, and *Pinus sylvestris*, to dry years in the West Carpathians, Slovakia. Utilizing data from ICP Forests Level I plots, we identified species-specific growth declines, particularly in *Pinus sylvestris* and *Fagus sylvatica*, with milder radial growth declines for *Quercus petraea* and *Picea abies*. *Abies alba* exhibited a growth peak in the mid-2000s, followed by a decline in the end of the observed period. Elevation emerged as the only significant environmental predictor, explaining 3.5% of growth variability during dry periods, suggesting a potential mitigating effect. The scope of this study was limited by the complex interplay of ecological factors that influence tree growth, which vary across the ICP Forests Level I monitoring sites. Nonetheless, our findings enhance the understanding of species-specific growth responses and offer insights for the climate-smart management of temperate forests under changing conditions.

## 1. Introduction

Tree species have evolved to be resilient and can recover from a variety of extreme climate-related events, thriving even under harsh environmental conditions. Nevertheless, each species has its own threshold for survival in such extreme conditions, which fundamentally shapes its geographical distribution. The observed climate change in the Central European region is characterized by increased intensity and frequency of extreme meteorological events, including droughts, windthrows, unseasonal frost, etc. [1,2]. Droughts led by heatwaves have caused serious environmental and economic issues, including wildfires and production losses in agriculture and forestry in the last decade, increasing the public need for drought risk management [3]. These findings utilized in simulations by the Intergovernmental Panel on Climate Change (IPCC) within their Special Report on Emission Scenarios (SRES A2, A1B and B1 scenarios) [4] indicate that the air temperature in the Western Carpathian region will rise, with 30-year averages increasing by approximately 2–4 °C by the century’s end. The modelled precipitation patterns are expected to vary considerably, but overall, an annual increase of roughly 10% is projected, with greater increments in the northern regions, and smaller ones in the southern regions [4]. The frequency of spring and summer droughts is projected to increase across Europe, including the Western Carpathian region, as a consequence of changes in precipitation spatio-temporal distribution patterns [5,6,7]. During the last decades, the 2003 millennium drought was often invoked as an example of a “hotter drought” [8] and was classified as the most severe event in Europe in the last 500 years. At a European scale, the 2003 drought remains associated with wide negative impacts on forest growth and productivity [9]. Similarly, the drought event in 2018 caused unprecedented drought-induced tree mortality in many species throughout the European region, with unexpectedly strong drought legacy effects detected in 2019 [10]. Currently, the summer of 2022, which followed a dry spring with above-average air temperatures, is assessed as the “hottest on record” in a large part of Europe [11]. Extraordinary heatwaves and droughts were recorded in Europe also in the years 2006, 2007, 2015, and 2019.

Future projections indicate drastic shifts in vegetation potentials in all parts of Europe [12]. Boreal forests could lose up to 75% of their current potential, while Mediterranean *Quercus* forests and steppes would double their potential areas [13]. On the other hand, published results indicate that oaks (*Quercus petraea* and *Quercus robur*), which are considered more resilient to extreme droughts, showed a growth decline when droughts occurred in spring [14]. In Central Europe, a serious decrease in the distribution of *Picea abies* Karst, which is one of the dominating species in the boreal as well as in the montane and altimontane forests, was revealed [12]. As for *Pinus sylvestris*, individuals that experienced more frequent droughts over a long period were less resistant to extreme droughts [14]. Different tree species, accordingly, often exhibit contrasting growth responses to climate change manifestations [15]. The rising evaporative demand (vapor pressure deficit) was indicated as a dominant environmental driver that constrains the growth of forest tree species in Western and Central Europe [16]. The growth rate in 2000–2004, compared with 1997–1999, showed reductions of up to 50% on half of the International Cooperative Programme on Assessment and Monitoring of Air Pollution Effects on Forests—ICP Forests—Level II plots examined in Italy [17]. In Western European forests, the growth changes during the period 1980–2007 strongly varied across tree species and ranged from +42% in mountain contexts to −17% in the Mediterranean context. Recent studies identified growth declines in *Q. pubescens* and, over the latest years, also in *Q. petraea* and *F. sylvatica* [18]. Based on a study [19] from mixed mountain forests in Europe, also the Norway spruce showed a significant decline in productivity (−26%) over the period 1980–2010, while the annual volume increment in fir rose significantly from 7.2 m^3^ha^−1^·y^−1^ to 11.3 m^3^· ha^−1^·y^−1^ (+36%).

Currently, an extensive discussion regarding the potential of tree species in future forests is continuing. Arguments are primarily based on niche requirements and species physiology [20,21,22,23]. The current models consider various climate scenarios, forest pest gradations, biological invasions, and other potential limitations [1,24]. Predicting how climate change impacts the health and productivity of tree species is challenging due to various potential disturbances and the dynamic nature of forest attributes, including species mixture, density, and structural diversity, which contribute to the non-linear patterns of the growth responses [25,26]. The potential of species under climate change likely represents a trade-off between the various negative and positive impacts and the related feedback that allows growth and regeneration. Recent findings indicated that the ability of forests tree species to withstand and recover from climatic extremes can be enhanced by management practices. These include assisted migration, early-stage thinning to decrease competition, and employing small-group selection harvests to preserve the characteristics typical of mature, older stands [25]. Moreover, the development of site-specific, climate-smart management approaches relies on a deep understanding of the principal trade-offs among climatic events, site characteristics, and species-specific growth responses.

Growth and increment, in general, are both usable predictors of the drought sensitivity of plant populations [27,28,29]. Long-term measurements of radial increments across diverse social and environmental conditions, especially under the current climate stressors, have the potential to reveal significant growth signals that underscore the adaptive capacities of individual species to changing conditions. In this paper, we used a wide dataset of radial increments measured according to ICP Forests protocols [30] and collected in the Western Carpathians, Slovakia, since 1988 to compare the populations of the most common tree species. We aimed to test two hypotheses:

**H1**. *The radial increment trends of tree species will display distinct patterns during the observed period, reflecting differential growth responses to drought*.

**H2**. *The decline in radial growth observed during dry years will appear to be exacerbated by elevation- and soil-related site conditions, which may influence species-specific growth patterns*.

## 2. Materials and Methods

### 2.1. ICP Forests Level I Tree Growth Database

Permanent tree radial increment monitoring in Slovakia has been established since 1988, with the primary purpose of monitoring the impacts of air pollution on the health of forests. Radial increment is measured according to the ICP Forests tree growth protocol [31]. The dataset comprises 167,717 annual circumference records from 1988 to 2023, measured in a total of 8133 individuals. We specifically used increment records of widespread Western Carpathian tree species such as common beech (*Fagus sylvatica* L.), Norway spruce (*Picea abies* Karst.), silver fir (*Abies alba* Mill.), sessile oak (*Quercus petraea* Matt.), and Scots pine (*Pinus sylvestris* L.) (Table 1).

### 2.2. Detrending and Representativeness

The ICP Forests Level I network in Slovakia encompasses 130 permanent monitoring plots arranged in a 16 × 16 km grid. The increment records utilized in the analysis were measured on 115 original plots (Figure 1), with the selection criterion being the presence of at least one individual of the target species (Table 1 and Appendix A, Appendix A). The spatial distribution of the increment records mirrored the distribution areas of the selected species in the Western Carpathians. To eliminate the effect of tree size and to enhance representativeness, we used the relative basal area increment (RBAI) calculated as the annual basal area increment relative to the basal area in the year of measurement. Detrending of the mean RBAI was conducted through the averaging of equitably distributed increment records in 5 cm circumference classes through bootstrap sampling. The number of circumference classes was individually set for each species, based on the variation in the dataset. To enhance the robustness of the results, we permitted repeated selection of the same samples. Each repeated sampling consisted in the selection and averaging of five increment records in each circumference class. Consequently, we plotted the increment timeline with 95% confidence intervals of the mean for each species. The representative mean RBAI for the species in each year was calculated by averaging 20-time-repeated bootstrap sampling from the dataset.

### 2.3. Site and Climatic Characteristics

As site characteristics, we used basic digital terrain model derivates such as elevation, slope, aspect (in 10 × 10 m resolution), tree species richness, and soil characteristics using the ICP Forests Level I soil survey database. We especially focused on pH (H_2_O), organic carbon and total nitrogen content, and physical soil characteristics such as clay, silt, and sand proportions at a 0–10 cm soil depth. The soil characteristics were investigated in 2006–2007 as part of the BioSoil Demonstration Project initiated during the Forest Focus-Scheme survey in 2008 [33].

For identifying the dry years, we used the grided 0.1° EOBS dataset of mean daily air temperature and precipitation totals [34]. We conducted a spatial interpolation of the climatic characteristics for each plot, using the weighted mean from the 4 nearest EOBS grid points with systematic corrections according to the elevation gradient (0.7 °C.100 m^−1^ for temperature and 40 mm.100 m^−1^ for precipitation). To test H2, we focused on summer droughts, which have been identified as important factors limiting tree growth [35]. We selected years with total summer precipitation (for June, July, and August) lower than normal (1950–2021) in more than 90% of the total ICP Forests Level I plots. In subsequent analyses, we closely examined tree growth in the years 1990, 1992, 1993, 1994, 2000, 2003, 2012, 2013, 2015, 2017, 2019, and 2022 (Figure 2), many of which were used as negative pointer years in other recent European studies [36].

### 2.4. Analysis

To test H1, we used the slope of the regression line of the plotted timeline (mean RBAI) to compare species production trends in R environment [37]. To analyze the statistical significance of the observed trends, we used the *p*-values of the linear regression model using the mean RBAI as the predicted variable and time (YEAR) as the predictor variable (Equation (1)).
RBAI(SP)∼YEAR(1)

To investigate the presence of non-linearity in growth trends, we applied the generalized additive model (GAM) using *mgcv* R package [38] with Gamma log link function and cubic regression spline smoothers (Equation (2)). As the distribution of the classes according to the diameter at breast height (DBH) was not homogeneous across the plots and to assess the difference in growth trends across tree-size gradients, we fitted the model for various temporal resolution values (1990–2021, 2000–2021 and 2010–2021) and for three different DBH sizes (20, 30, and 40 cm). Thus, a fitted curve was obtained representing the basal area increment not for individual trees but for cohorts of trees that reached a certain and constant DBH in every year. To show the variation of the estimate, we calculated 95% confidence intervals.
BAI~te (DBH_real, year, k = 4, bs = “cs”)(2)

For H2, we conducted a regression of environmental variables with principal components—PCA regression [39] model—for each plot and species to determine the main drivers of the RBAI in the selected dry years. A multivariate approach could reveal the most important species-specific predictors and explain the variation in drought-limited growth between sites and species. To highlight the species-specific growth under harsh conditions, we calculated centroids according to the PCA scores and ellipses as the 95th percentile of PC1 and PC2 coordinates.

## 3. Results

### 3.1. Comparing Three Decades of Species Growth (H1)

Dry years resulted in a decrease in the mean RBAI across species. With the exception of silver fir, all species exhibited negative trends (Figure 3). The steepest negative slope was observed for beech. The non-significant trend observed for silver fir was a consequence of the RBAI peak in the mid-2000s. Currently, we are observing a decrease in the increment comparable to that recorded in the early 1990s. After a consistent decline and minor fluctuations in the mean RBAI, spruce displayed irregular growth patterns in 2021–2022 but showed signs of growth recovery in 2023.

According to the GAM model (Figure 4), the increment distribution varied within species based on social and DBH classes. This revealed that for all species, except pine, the basal area increment for smaller individuals was smoother than for larger ones. The difference in absolute basal area increments between the DBH classes was most pronounced for silver fir and spruce. Decreasing trends were observed across the DBH gradient within the species.

### 3.2. Comparing Radial Growth in Non-Favourable Climatic Conditions (H2) and Limiting Factors

PCA ordination showed differences in site-specific species growth reactions to selected dry years (Figure 2). The growth reactions (RBAI) in dry years, based on the position of the centroids, differed along both PCA axes. The centroid of silver fir, a less–sensitive species with a relatively high RBAI throughout the whole period, was situated in the lower left part of the ordination, closer to that of oak compared to those of the other conifers, which showed higher decline rates. The variation in species reactions according to ellipses (Figure 5) was the highest for beech and silver fir. On the other hand, the least variable growth reactions were observed for spruce and pine.

According to the PCA regression, elevation was the only significant predictor, accounting for 3.5% of the variability in the RBAI along both the PC1 and the PC2 axes (Table 2). Other variables that were almost significant (*p* < 0.1), namely, soil pH, total nitrogen content, and mean annual temperature, explained 5.8% of the RBAI variability. The most pronounced pattern in the ordination represented climate-induced site conditions along the altitudinal gradient. Despite the high observed variability, during the dry years, the best growth for silver fir was observed at sites at high elevations. Conversely, the species consistently displayed a lower RBAI on nitrophilous sites with higher pH (H_2_O) and clay content.

## 4. Discussion

### 4.1. Observed Growth Trends (H1)

Our analytical approach, using detrendization through bootstrapping of the extensive and harmonized ICP Forests Level I dataset is less affected by survivorship bias compared the analysis of tree ring data. The trend analysis (Figure 3) revealed various, mostly significant decreasing tendencies in the radial growth of the investigated tree species over the last three decades. Our results confirmed an on-average decrease in tree species growth, which was already pointed out by several authors [26,40]. The production decrease and increased mortality of trees in the Central European region were often linked to drought events in the last decades, particularly in 2000, 2003 [41], and 2022. In the case of conifers, when considering the centroid positions (Figure 5), it was observed that the RBAI development was similar in pine and spruce during dry years and differed from that of fir, whose production was less affected by the drought-induced growth decline. The decline in radial growth is linked to the carbon balance of forest ecosystems and, consequently, to their overall ecological integrity [42,43,44]. Besides reflecting tree health, radial growth patterns are also predictive of a species’ competitive abilities and adaptive responses to changing environmental conditions, such as those imposed by drought. For example, the less pronounced decline in radial growth in *Quercus petraea*, compared to other deciduous species, may signal a competitive advantage in resilience to drought stress. Findings from other studies [45], linked the declining radial growth patterns of 36 species to tree mortality. The interpretation of radial growth measurements as indicators of a species’ adaptability is challenging due to the difficulty in distinguishing whether a decline in growth is an adaptive mechanism, e.g., to avert embolism through vessel width reduction, or a sign of stress-related damage [46]. Consequently, to conclusively determine the species-specific resilience to environmental stressors, analyses of radial growth declines should be complemented by mortality rate data.

#### 4.1.1. *Abies alba*

Successful silver fir growth and regeneration were already described even from the Mediterranean region, and several authors attribute a high future potential to this species [47,48,49]. In accordance with previous research [26,50], it is noteworthy that the fir population in Europe has shown a higher resistance to drought events, even after facing the adverse effects of SO_2_ and NOx pollution in the latter half of the 20th century. However, based on our findings (Figure 3), we conclude that since the year 2000, the frequency and nature of drought events have once again led to a decline in this species vitality. Despite the observed trend turnover, the values of RBAI still varied between 2 and 3%, decreasing under 2% only in extreme dry seasons (Figure 3). Recent research reported concerns about silver fir mortality as a consequence of increased evaporative demand [51]. The variation in site-specific RBAI development in dry years was the most significant among those of the examined conifers and was shifted downwards in the ordination space (Figure 5), indicating higher RBAI values in general.

#### 4.1.2. *Picea abies*

The observed non-significant RBAI decrease for spruce may be attributed to increased mortality (as evidenced in [47,52]), which typically occurs after extremely dry years. Such mortality can introduce a bias into the measurements, as the surviving trees could be less limited by water resources and competition. Notably, when examining the variation in increment during dry years (Figure 5), it becomes evident that the variation for this species was the lowest despite the wide variety of sites studied and shifted towards the right side (indicating a lower RBAI) of the ordination space. In comparison to other tree species, the RBAI values for spruce remained consistently below 2% throughout the entire observation period.

#### 4.1.3. *Pinus sylvestris*

The significant decrease in the pine RBAI in the Western Carpathians confirmed the indications of a decline in pine growth across the broader Central European region [53,54]. The Scots pine distribution area in the Western Carpathians is mostly connected to dryer sandy or calcareous soil conditions, where other species are less competitive. Even despite high physiological resistance and adaptation capacity, the continuous growth decline observed after the droughts of 2000–2003 raises questions about whether the extreme sites preferred by Scots pine will remain suitable for forest regeneration and growth in the near future. Our observations showed that in last three decades, the RBAI approached values around 1% (Figure 3), while in the PCA ordination (Figure 5), the variation appeared shifted to the right side compared to those of the other investigated species (indicating the lowest RBAI).

#### 4.1.4. *Fagus sylvatica*

The observed decreasing trend for common beech (Figure 3) was highly significant, even when compared to the data from the other investigated deciduous species. The sensitivity of beech to drought is well known [36,55,56,57]. Our findings indicated that after the droughts in 2000 and 2003, the RBAI decreased to a value under 2% and has not recovered yet. Other research works with longer observation period dated the beech decline to the 1980s [58]. Based on the PCA (Figure 5), the highest variation in increment was recorded during dry years, reflecting the wide and heterogeneous site conditions within the beech distribution area in the Western Carpathians. A concerning sign, particularly when considering the importance of beech-dominated forests in the carbon balance of European forests [57], is the steep decline in the RBAI over the observed three decades. Despite the production decrease, the regeneration potential and competitiveness of beech still seem to be high [59,60].

#### 4.1.5. *Quercus petraea*

In the case of sessile oak, which showed a less significant decreasing trend (*p* < 0.01), we observed a less steep decrease in the RBAI, currently varying around 2%. Compared to those of other deciduous species, the variation in oak site-specific RBAI development (Figure 5) was the lowest. Focusing on the centroids, the oak RBAI in dry years was between those of the sensitive beech and the more resistant silver fir. Our results align with already published [61,62] findings about the higher resistance to drought of this species, compared to other deciduous broadleaved tree species. Lower oak drought-induced growth decline patterns were predicted also through growth simulators [63].

### 4.2. Adaptation Potential and Limiting Factors (H2)

Despite the use of classic stem circumference measurements according to ICP Forests Tree Growth protocols [31] (rather than the more precise dendrochronology methods), all investigated species exhibited a relatively synchronous growth decline in selected dry years, in contrast to an increase in RBAI during normal seasons (Figure 3). Repeated mechanical measurements of the stem circumference, even at the permanently marked point, can suffer from various errors, including observer bias, consequences of bark damage, or daily stem size fluctuations. Despite the presence of errors, annual time series provide valuable information about species’ growth trends. The presence of dendroecological signals emphasizes the importance of conducting exploratory analyses, such as PCA analysis. It is worth noting that our study design, based on ICP Forests Level I plots and field measurements, is not specifically tailored for the reconstruction of ecological gradients. Consequently, it involves varying levels of captured variability among individual predictors, which could result in lower values of the explained variance in the RBAI. The presented results showcased the in situ growth patterns of various species, accounting for the impact of unequal age and individual size distributions, while preserving all site-specific growth variations. Given the limitations and features of the study design and the RBAI as a relative variable, interpretations should consider that, especially in PCA, the site positions in the ordination are influenced also by other factors. Management, competition, and other dynamic forest structural parameters that significantly impact the RBAI were not considered in our analysis. These limitations represent one of the reasons for the low explained variability along the PCA axes. Despite the limited explained variability and the lack of statistical significance for the majority of the observed environmental gradients, we utilized the results of the exploratory Principal Component Analysis (PCA) to identify potentially important site characteristics.

Species behavior was investigated through radial growth measurements in 12 dry years. The PCA revealed two main gradients in radial increment variation across the sites. The first gradient presented the varying behaviors of tree species under drought conditions along their vertical distribution. We concluded that strongly correlated variables (elevation, annual precipitation, and mean annual temperature) can be collectively considered as one factor (Appendix A) explaining 8.13% of the variability in the RBAI along both PCA axes. Especially fir and oak showed a minor difference between the minimum and the maximum site–specific RBAI values, whereas no difference was observed for the remaining species in the elevation gradient (Figure 5). Other sources addressing growth reactions along elevation gradients pointed to varying and seasonally changing growth patterns (wet vs. dry seasons) [35]. Published research showed regional differences or deviations from the expected growth at higher elevations [64,65,66] or better water use efficiency that can lead to more resilient populations in mountainous regions [67]. In the case of beech, our findings did not clearly demonstrate higher drought resistance at higher elevation, as observed in Germany [58]. Our results showed a growth decline at middle elevations, despite some sites at lower altitudes exhibiting the highest growth rates. This, in fact, shifted the beech centroid to the upper left quadrant of the ordination (Figure 5). Such patterns could be caused by the age structure and the presence of a calcareous bedrock at middle and low elevations. In contrast, the best growth rates during dry years were observed in oak populations at high elevations. In the case of conifers, only silver fir showed a slightly better growth at high elevations, which aligns with published research that described the most productive fir sites [68]. Based on our findings, we can conclude that we concur with previously published research [69] and, due to minor differences, we do not expect compensation for the more pronounced drought-related decline at lower elevations in the Western Carpathian region. Furthermore, it is important to note that the presented growth decline in tree species populations cannot be simply scaled up to the forest ecosystem level. At this level, processes such as tree regeneration, mortality, respiration, and nutrient cycling are crucial for forest production and carbon uptake.

The RBAI variability along the PC2 axis was slightly and insignificantly explained by the gradients of soil pH and clay content (Figure 5, Table 2). Compared to Cambisols developed from sandstone, metamorphic, or volcanic parent material, soils originating from claystone or limestone have a higher clay content. These soils exhibit an increased capacity for water retention. However, they can also exhibit greater resistance to water uptake, which can be limiting during extreme or long-term droughts [47]. Water uptake is primarily influenced by the distribution of fine roots, which is shaped by the soil physical characteristics. Heavy soils are often characterized by the highest fine root density in the upper soil layers, which is responsible for the majority of water uptake. In dry seasons, there is evidence that the lack of uptake from the upper layers cannot be efficiently compensated by the water uptake from deeper layers [70]. In the case of limestone-derived soils, the general water balance tends to be more drought-vulnerable as a consequence of bedrock permeability. Considering the PCA results (Figure 5), sites with the lowest RBAI were consistently located in the right side of the ordination, shifted in the direction of higher soil pH (H_2_O) values and higher clay content. This pointed to a slight strengthening effect on a growth decline during the dry years. Compared to the predictions of process-based models under Representative Concentration Pathway (RCP) scenarios [71], our empirical evidence showed an opposite impact of clay on growth during dry periods.

## 5. Conclusions

Over the course of three decades, the ICP Forests Level I monitoring in the Western Carpathians captured a mostly significant and consistent decline in the growth of five key tree species. As a result of bootstrap sampling-based detrendization, the observed decline in radial growth was primarily attributed to drought events. We conclude that Hypothesis 1 was confirmed, as different tree species exhibited distinct growth trends as well as experienced growth declines during dry years. *Abies alba*, which had previously demonstrated resilience and recovery from SO_2_ damage, showed a declining trend since the year 2000. Conversely, *Picea abies* exhibited a non-significant decrease in radial growth, characterized by a low spatial RBAI variation during dry years. Among conifers, *Pinus sylvestris* experienced the most significant decline in growth. Within the group of deciduous trees, *Fagus sylvatica* stood out as being highly susceptible to drought, with a significant decline in growth observed since the 1990s. In contrast, *Quercus petraea* showed a less pronounced decline, indicating higher resistance to drought when compared to other deciduous tree species. In assessing the growth responses of these species to drought along elevation gradients, our findings suggest only a minor effect of higher altitudes on drought sensitivity, notably, in *Abies alba* and *Quercus petraea*. The majority of the species exhibited no reaction to the elevation gradient. Furthermore, while only partially confirming Hypothesis H2, it is important to highlight the consistent patterns, suggested by the soil characteristics, of drought-induced growth decline, despite the lack of statistical significance. Soils with high pH and clay content or those on a limestone bedrock might have contributed to this decline. While clay soils have the ability to retain water, they can hinder water uptake during extreme drought periods, intensifying the overall impact of drought. Additionally, early dry-out of the soil can result from bedrock permeability. These findings improve our comprehensive understanding of the intricate dynamics of tree growth in response to evolving in situ environmental conditions. They also underscore the importance of examining species-specific responses and the effects of elevation, mixture, and soil gradients within the broader context of forest tree species and ecosystem resilience. This study highlights the urgent need for site-specific climate-smart management strategies to improve forest resilience.

## Figures and Tables

**Figure 1 plants-12-04081-f001:**
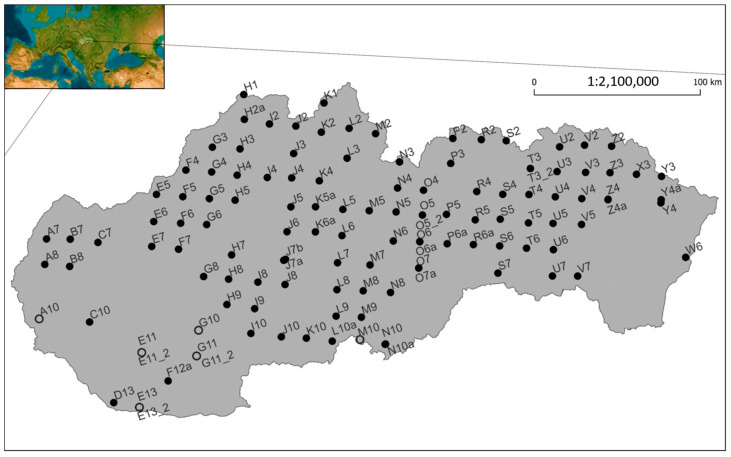
Map of active ICP Forests Level I plots and plots included into the analysis. Black color indicates plots with the occurrence of the investigated species included into the analysis; every plot is marked by an ID that consists of transect and serial number (A7). Plots that were relocated to areas with similar conditions nearby the original ones due to various reasons are denoted by lowercase letters (a, b, etc.), while plots that were reestablished in the exact same location are marked with ‘_2’.

**Figure 2 plants-12-04081-f002:**
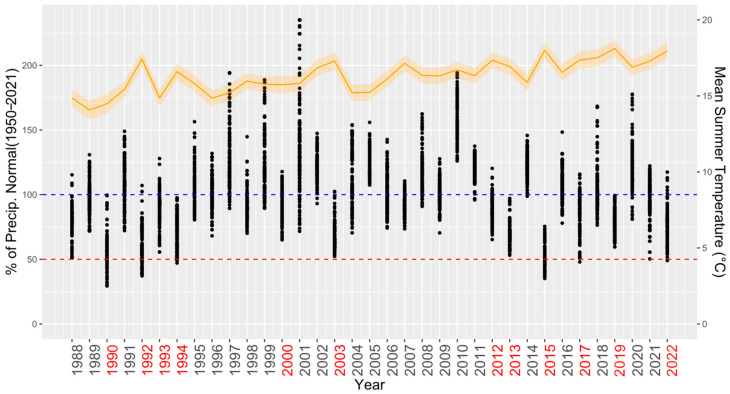
Selected dry years (red) based on summer precipitation compared to normal precipitation levels (1950–2021). Dots represents annual values for each plot. Blue dashed line indicates 100%, and red dashed line indicates 50% of long-term normal precipitation. Orange line represents mean summer temperatures with 95% confidence intervals (orange shaded area) for the means derived from all ICP Forests Level I plots.

**Figure 3 plants-12-04081-f003:**
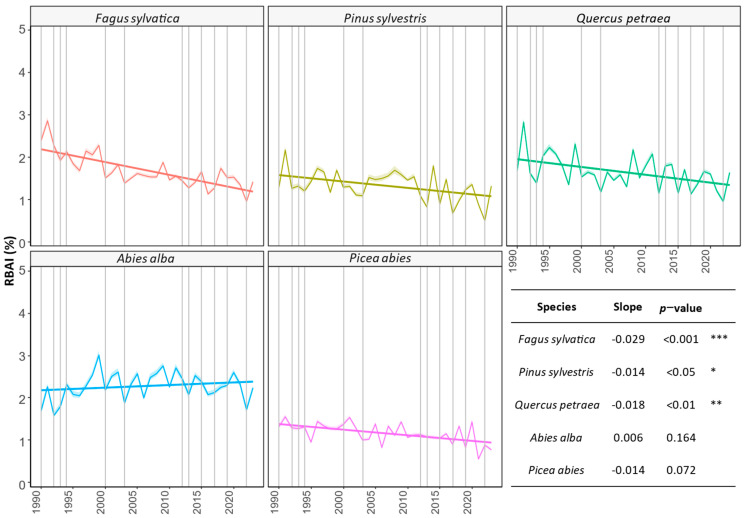
Growth trends of selected species during the observed period (1989–2023) expressed by coloured linear regression lines for each species. Gray vertical lines point to selected dry years according to Figure 2. The observed trends in mean RBAI were significant for *Fagus sylvatica*, *Pinus sylvestris*, and *Quercus petraea*. The RBAI decrease during the observed period was the highest for *Fagus sylvatica* (−0.029). Asterisks represent statistical significance of the trends (* = *p* < 0.05, ** = *p* < 0.01, *** = *p* < 0.001).

**Figure 4 plants-12-04081-f004:**
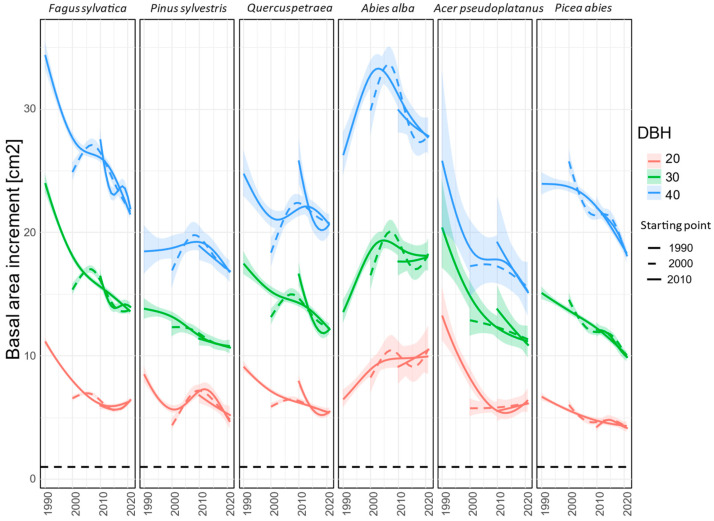
Development of BAI for three different DBH values (20, 30, 40) and different periods (1990–2021, 2000–2021, and 2010–2021). Ribbons around growth trend curves represent the 95% confidence interval.

**Figure 5 plants-12-04081-f005:**
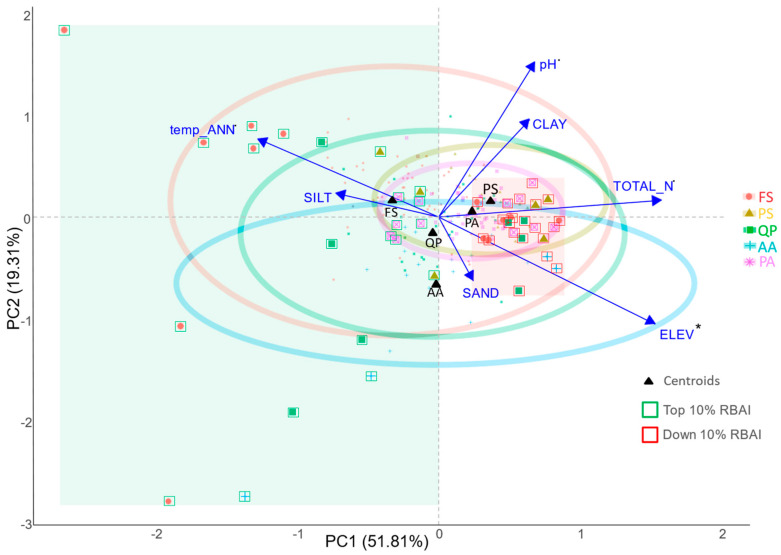
PCA plot of the RBAI in dry years. The position of each point (shaped and coloured according to the species) represent the development of the RBAI in 12 selected dry years. Black triangles represent centroids calculated from PCA scores of species from all plots. Centroids indicate the average position for each species (FS = *Fagus sylvatica*, PS = *Pinus sylvestris*, QP = *Quercus petraea*, AA = *Abies alba*, PA = *Picea abies*) in multidimensional space. Highlighted ellipsoids represent 95% of species-specific variation in the RBAI on different sites through 12 dry years. PC1 axis explained 51.81% of variability, and PC2 explained 19.31% of variability in the values of the RBAI in dry years. Length and angle of the vectors of the plotted environmental variables represent the strength and character of the relationship with the PCA axes. Highlighted quadrants represent the variation between the best-growing and the worst-growing sites. Coloured squares are used to zoom in on and highlight the 10% of plots that experienced the highest (green) and the lowest (red) growth increments during the 12 dry years.temp_ANN = mean annual temperature, SILT, SAND, CLAY = proportion of soil fractions at a 0–10 cm soil depth, TOTAL_N = total nitrogen content at a 0–10 cm soil depth, pH = pH of H_2_O, ELEV= elevation. Asterisks represents statistical significance of environmental variables (. *p* < 0.1, * *p* < 0.05).

**Table 1 plants-12-04081-t001:** Number of trees and increment records for each species in comparison to species proportion in Slovakia according to the national forest inventory (NFI [32]).

Species	Species Proportion [% ± SD] (NFI)	Number of Plots (ICP Forests I)	Number of Individuals (ICP Forests I)	Number of Increment Records (ICP Forests I)
*Fagus sylvatica*	30.1 ± 2.4	67	2251	53,255
*Picea abies*	18.7 ± 2.0	55	2236	43,438
*Quercus petraea*	7.2 ± 1.3	34	651	14,703
*Pinus sylvestris*	4.8 ± 1.1	27	643	14,117
*Abies alba*	2.8 ± 0.9	19	351	7485
Σ	63.6	115 orig. plots	6132	132,998

**Table 2 plants-12-04081-t002:** Results of the PCA regression (TREE RICHNESS = tree species richness, CLAY, SILT, SAND = proportion of soil fractions at a 0–10 cm depth, TOTAL_N = total nitrogen content at a 0–10 cm soil depth, ORGANIC_C = organic carbon content at a 0–10 cm soil depth, C_N_ratio = ratio between organic carbon and total nitrogen contents at a 0–10 cm soil depth, precip_ANN= annual precipitation, temp_ANN = mean annual temperature, R^2^ = explained variance, *p*-value = statistical signifcance (. *p* < 0.1, * *p* < 0.05).

Site Characteristics	R^2^	*p*-Value	
ELEVATION	3.45	0.028	*
ASPECT	0.75	0.493	
SLOPE	0.22	0.808	
TREE RICHNESS	1.53	0.205	
pH (H_2_O)	2.79	0.058	.
CLAY	1.34	0.279	
SILT	0.58	0.564	
SAND	0.45	0.657	
TOTAL_N	2.51	0.082	.
ORGANIC_C	1.84	0.151	
C_N_ratio	0.23	0.801	
precip_ANN	2.47	0.095	.
temp_ANN	2.21	0.120	

## Data Availability

All data are available and can be produced upon request. The data are not publicly available due to ICP Forests Intellectual Property and Publication Policy.

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
