# Peer review of "Declining Radial Growth in Major Western Carpathian Tree Species: Insights from Three Decades of Temperate Forest Monitoring"

_plants, 2023, doi:10.3390/plants12244081_

Round 1

Reviewer 1 Report

Comments and Suggestions for Authors

Comments for Author,

I have read your paper carefully and I have found that your paper is interesting. However, I have some minor comments before accepted this paper. 

Specific comments:

Please rephrase these below sentences

"Forest tree species are well adapted to overcome various combinations of climatic  events and successfully regenerate, even in extreme site conditions. However, the ecological tolerance of each species to such extremes is limited, and these limitations are the main  drivers of species' distribution areas."

"Predicting the effects of climate change on tree species' health and  productivity is complex due to potential interplay with disturbance occurrences and the  dynamics of forest characteristics, such as species composition, density, and age-related  variation that are reasons of non-linearity in growth response patterns [23,24]."

H1: Species exhibit different adaptation potential, as measured by increment trends during observed period. 

H2: The sensitivity of species, as evidenced by changes in radial increment during dry years, varies across different site conditions. 

Even the results part is interesting, the discussion part is superficial. So, this part is needed to detail.

Finally, the conclusion remark is missed. Please add your future recommandation.

Best Regards

Reviewer 2 Report

Comments and Suggestions for Authors

In this manuscript (plants-2722065) entitled "Declining radial growth in major Western Carpathian tree species: insights from three decades of temperate forest monitoring" submitted to Plants, Jergus Rybar and colleagues have examined radial growth patterns of five key European temperate forest tree species: Fagus sylvatica L., Picea abies Karst., Abies alba Mill., Quercus petraea Matt., and Pinus sylvestris L. We ex-plored their responses to extreme climatic events, notably droughts, using data from ICP Forests Level I plots in the West Carpathians, Slovakia. This research is interesting and convincing, but minor points need to be addressed to improve the quality of this manuscript.

1. To better understand this study, representative growth pictures of forest tree analyzed here should be shown in the revised manuscript.

2. For Figures 1, abbreviations A-Z presented in this Figure should be explained in the revised legand.

3. For Figures 2, high temperature usually come together with drought. Authors should consider to show the temperature data in the revised Figure 2.

4, Full name of the abbreviation ICP, PCA, IPCC SRES A2, A1B, B1, and DBH presented here should be spelt out at their first appearance in the revised manuscript.

Reviewer 3 Report

Comments and Suggestions for Authors

The experiment conducted by the authors broadens the knowledge on the “Declining radial growth in major Western Carpathian tree species: insights from three decades of temperate forest monitoring”. There are some comments that should be taken into account by authors, which I believe are significant and important aspects that need to be thoroughly addressed in authors revision.

The main concern is:

(1) The abstract doesn't provide a clear problem statement or research gap. There is no mention of potential limitations or challenges. To improve this abstract, consider condensing the content for better clarity and conciseness. Clearly state the research objectives and the significance of the study.

(2) Moreover, the presentation of the results in the abstract should be carefully and completely revised.

(3) The introduction is ok however; the hypothesis is not clear. So, at the end of this section, please illustrate what hypothesis this investigation aimed to test.

(4) Authors should discuss at the end of the discussion section how their results fill the gap of previous studies.

(5) Some papers should be added to enhance discussion part.

(6) Throughout the manuscript, there is also a lack of indication of what is innovative in this paper and what the authors have contributed to the current state of knowledge.

(7) The conclusion section must be rewritten. Authors should include specific results of their research, which extend the current state of knowledge. 

(8) References need to be cross-checked.

(9) Minor corrections in respect of spellings and some places in grammar.

Comments on the Quality of English Language

Minor corrections in respect of spellings and some places in grammar.

Reviewer 4 Report

Comments and Suggestions for Authors

Major remarks:

1) I have major concerns about your hypotheses:
H1: In my opinion the increment trend cannot be regarded as indication of an adaptation to drought. Of course, the tree species are responding to drought with reduced growth but why is it an adaptation?

H2: A hypothesis should express a clear statement which either can be confirmed or rejected.  That the sensitivity of species “vary” according to site conditions is a very general statement which nearly always will be true.
Furtheron, I do not agree that radial increment shows sensitivity to drought. Less RBAI can also be an indication of higher drought tolerance by avoiding hydraulic failure (isohydric species). Only mortality rates in dry years from the different plots could allow a statement to the drought sensitivity of the species. Do have observations about the mortality at the level I plots? That would be much more decisive for the drought sensitivity of the species as radial growth.

Please, strengthen the hypotheses.

2) Abies alba: there is a lower number of individuals included in the study compared to other species. Can you proof that the individuals of AA are distributed along the total elevation gradient or are they  preferentially growing in higher elevations with higher precipitation which would explain the lacking growth reduction?

3) By relating the increment trends to environmental factors I think, the restriction of soil traits to the upper 10 cm of the mineral soil are a severe limitation. It is not sufficient for species with a deeper main rooting zone (as PS, QP, AA and FS) to mention only the very topsoil layer.

4) The introduction could be shorter because vegetation shifts are mentioned as well as overall drought reponses which are however not leading to the hypotheses and results and are not subject of the study.

5) Drought and heatwaves are always mentioned in the text in combination: Do you have evidence that heat could restrict radial growth by itself or only by causing/increasing the drought stress?

6) In my opinion, wood production is an economic feature and not an ecological one. Survival and fitness of a species does not depend on radial growth in dry years. See comment 1) regarding lacking information about mortality.

7) The effects of species mixtures are not mentioned at all or did I have not seen it? Combinations of different mixtures could influence the drought response a lot.

8) Only Quercus petraea is shown in the data set. I wonder, if it is not mixed with Quercus robur or can you exclude that it is growing on the plots, too.

Minor remarks, more detailed

1) page 1, introduction, line 36: “increase of precipitation by 10 %” but in line 37/38 “more frequent droughts”, how is it fitting together.

2) page 2, line 54: please, add a citation.

3) page 2, line 70: ha-1 instead just ha.

4) page 2, line 74: sentence incomplete: “…are considering”.

5) page 2, line 88: Only from an economic point of view growth is showing “climate (better: drought) sensitivity” (see above).

6) page 3, line 114/115: sentence unclear.

7) page 7, fig. 4, FS and PS are commented to show “marked growth reductions” whereas QP and PA have “a mild negative trend”. For me, PA shows a stronger reaction as PS, or am I wrong?

8) Fig. 5: The colours and symbols are very difficult to identify and to distinguish. Perhaps you find a better solution.

9) page 8, line 221/222: The site factors are not independent. Did you test on autocorrelations?

10) page 9, line 245. Even if it is in the title of this special issue, I normally avoid the word “vitality” because it is so unspecific. Please, make clear, what you want to express with it.

11) page 9, line 251/252. Here mortality is mentioned, but no results shown before (see above). Mortality however is not only relevant for spruce but also for the other species, isn’t it?

12) page 9, line 269: “shift of PS to the right side” I hardly can see in fig. 5.

13) page 10, line 273: vulnerability is not to be indicated by radial growth( see above).

14) page 10, line275/276: no recovery yet (not only in FS but also in other species except AA).

15) page 10, line 310: Better PCA axes to avoid confusion because fig. 5 always is mentioned as PCA graph.

16) page 10, line 317: The evident (auto)correlation of the factors is not shown.

17)references page 13, line 429: not capital letters in author names.

Reviewer 5 Report

Comments and Suggestions for Authors

I suggest to comment on the uncertainties or differences of repeated diameter measurements (carried out in ICP1 plots) as compared with tree-ring data. This could be added in line 294 of the ms.

I attach a commented pdf.

Comments on the Quality of English Language

I attach a commented pdf with suggested changes.

Round 2

Reviewer 3 Report

Comments and Suggestions for Authors

The comments have been mostly addressed. The manuscript has improved and can be published.